# Numerical behavior of NVIDIA tensor cores



Massimiliano Fasi[1], Nicholas J. Higham[2], Mantas Mikaitis[2] and Srikara Pranesh[2]

[1] School of Science and Technology, Örebro University, Örebro, Sweden
[2] Department of Mathematics, University of Manchester, Manchester, UK

## ABSTRACT

We explore the floating-point arithmetic implemented in the NVIDIA tensor cores, which are hardware accelerators for mixed-precision matrix multiplication available on the Volta, Turing, and Ampere microarchitectures. Using Volta V100, Turing T4, and Ampere A100 graphics cards, we determine what precision is used for the intermediate results, whether subnormal numbers are supported, what rounding mode is used, in which order the operations underlying the matrix multiplication are performed, and whether partial sums are normalized. These aspects are not documented by NVIDIA, and we gain insight by running carefully designed numerical experiments on these hardware units. Knowing the answers to these questions is important if one wishes to: (1) accurately simulate NVIDIA tensor cores on conventional hardware; (2) understand the differences between results produced by code that utilizes tensor cores and code that uses only IEEE 754-compliant arithmetic operations; and (3) build custom hardware whose behavior matches that of NVIDIA tensor cores. As part of this work we provide a test suite that can be easily adapted to test newer versions of the NVIDIA tensor cores as well as similar accelerators from other vendors, as they become available. Moreover, we identify a non-monotonicity issue affecting floating point multi-operand adders if the intermediate results are not normalized after each step.

## INTRODUCTION

One hundred and sixteen of the computers in the November 2020 TOP500 list (https://www.top500.org/lists/top500/2020/11/) are equipped with NVIDIA graphics processing units (GPUs) based on the Volta, Turing, and Ampere microarchitectures. A prominent feature of these GPUs is the *tensor cores*, which are specialized hardware accelerators for performing a matrix multiply-accumulate operation. Here, in particular, we focus on the tensor cores available on the NVIDIA V100 (Volta microarchitecture), T4 (Turing architecture), and A100 (Ampere architecture) GPUs, which implement the operation

Corresponding author
Mantas Mikaitis,
mantas.mikaitis@manchester.ac.uk

**Table 1 Processing units or architectures equipped with mixed-precision matrix multiply-accumulate accelerators.** The notation $m \times k \times n$ refers to the matrix multiply-accumulate operation in (1) where $C$ and $D$ are $m \times n$ matrices, and $A$ and $B$ have size $m \times k$ and $k \times n$, respectively.

| Year of release | Device | Matrix dimensions | Input format | Output format | References |
|---|---|---|---|---|---|
| 2016 | Google TPU v2 | $128 \times 128 \times 128$ | bfloat16 | binary32 | *Google (2020)* |
| 2017 | Google TPU v3 | $128 \times 128 \times 128$ | bfloat16 | binary32 | *Google (2020)* |
| 2017 | NVIDIA V100 | $4 \times 4 \times 4$ | binary16 | binary32 | *NVIDIA (2017)* |
| 2018 | NVIDIA T4 | $4 \times 4 \times 4$ | binary16 | binary32 | *NVIDIA (2018)* |
| 2019 | Arm v8.6-A | $2 \times 4 \times 2$ | bfloat16 | binary32 | *Arm Ltd (2020)* |
| 2020 | NVIDIA A100 | $8 \times 8 \times 4$ | bfloat16 | binary32 | *NVIDIA (2020b)* |
| | | $8 \times 8 \times 4$ | binary16 | binary32 | |
| | | $4 \times 8 \times 4$ | TensorFloat-32 | binary32 | |
| | | $4 \times 2 \times 2$ | binary64 | binary64 | |

$$
\underbrace{D}_{\substack{\text{binary16 or} \\ \text{binary32}}} = \underbrace{C}_{\substack{\text{binary16 or} \\ \text{binary32}}} + \underbrace{A}_{\text{binary16}} \times \underbrace{B}_{\text{binary16}}, \tag{1}
$$

where $A$, $B$, $C$, and $D$ are $4 \times 4$ matrices and "binaryxy" denotes the xy-bit format from the IEEE Standard 754 for floating-point arithmetic (*IEEE, 2019*). The entries of $A$ and $B$ must be in binary16 format, whereas those of $C$ and $D$ can be either binary16 or binary32 floating-point numbers depending on the accumulation mode. The newer A100 (Ampere microarchitecture) GPUs support a wider range of matrix dimensions, as well as an additional floating-point format, as shown in Table 1. The element $d_{ij}$ in (1) can be seen as the sum of $c_{ij}$ and the dot product between the $i$th row of $A$ and the $j$th column of $B$, so that, for instance

$$
d_{11} = a_{11}b_{11} + a_{12}b_{21} + a_{13}b_{31} + a_{14}b_{41} + c_{11}. \tag{2}
$$

Unfortunately, NVIDIA provides very little information about the numerical features of these units, and many questions naturally arise. The white paper that describes the Volta microarchitecture (*NVIDIA, 2017*, p. 15) states that[1]

*Tensor Cores operate on FP16 input data with FP32 accumulation. The FP16 multiply results in a full precision product that is then accumulated using FP32 addition with the other intermediate products for a $4 \times 4 \times 4$ matrix multiply.*

The official documentation (*NVIDIA, 2020a*) adds only a few more details:

*Element-wise multiplication of matrix A and B is performed with at least single precision. When .ctype or .dtype is .f32, accumulation of the intermediate values is performed with at least single precision. When both .ctype and .dtype are specified as .f16, the accumulation*

[1] The binary16 and binary32 formats are sometimes referred to as fp16 (or FP16) and fp32 (or FP32), respectively.

*is performed with at least half precision. The accumulation order, rounding and handling of subnormal inputs is unspecified.*

From a numerical point of view, many essential aspects of tensor cores are not specified. This lack of key information makes it challenging to simulate tensor cores on conventional IEEE 754-compliant systems and to build hardware that can reproduce their behavior. Moreover, it can lead to unexpected differences between the results computed on NVIDIA devices with tensor cores enabled or disabled.

We now briefly recall some key aspects of IEEE-compliant floating-point systems and the definitions and main properties of *normalization* and *subnormal numbers*, which are two important concepts in this work. A binary floating-point number $x$ has the form $(-1)^s \times m \times 2^{e-p}$, where $s$ is the sign bit, $p$ is the precision, $m \in [0, 2^p - 1]$ is the integer significand, and $e \in [e_{\min}, e_{\max}]$, with $e_{\min} = 1 - e_{\max}$, is the integer exponent. In order for $x$ to have a unique representation, the number system is *normalized* so that the most significant bit of $m$—the *implicit bit* in IEEE 754 parlance—is always set to 1 if $|x| \geq 2^{e_{\min}}$. Therefore, all floating-point numbers with $m \geq 2^{p-1}$ are normalized. Numbers below the smallest normalized number $2^{e_{\min}}$ in absolute value are called *subnormal numbers*, and are such that $e = e_{\min}$ and $0 < m < 2^{p-1}$. Subnormal numbers provide a means to represent values in the subnormal range, that is, between 0 and the minimum normalized number. They have lower precision than normalized values (from $p - 1$ bits to as low as 1 bit), and require special treatment to be implemented in software and hardware. Therefore it is not uncommon for hardware manufacturers not to support them, in order to avoid performance or chip area overheads.

In implementing floating-point operations the result of an operation must be normalized (if possible) by shifting the significand left or right until it falls within the interval $[2^{p-1}, 2^p - 1]$ and adjusting the exponent accordingly. More details can be found in *Muller et al. (2018*, Sec. 7.3).

The IEEE 754 standard for floating-point arithmetic provides a somewhat relaxed set of requirements for reduction operations such as dot product and multi-operand addition (*IEEE, 2019*, Sec. 9.4): the order in which the partial sums should be evaluated is not prescribed, and the use of a higher-precision internal format is allowed. In particular, the standard does not specify: (1) whether this internal format should be normalized, as it would be if the multi-operand addition were implemented using IEEE 754 elementary arithmetic operations, (2) which rounding mode should be used, and (3) when the rounding should happen. These loose requirements can potentially cause the results computed with a given multi-operand addition unit to be significantly different from those obtained using other hardware implementations or a software implementation based on IEEE 754-compliant elementary arithmetic operations.

With matrix multiplication being ubiquitous in artificial intelligence, accelerators for mixed-precision matrix multiply-accumulate operations are becoming widely available, as Table 1 shows. Hardware vendors often design these units focusing on performance rather than numerical reliability, and this may lead to the implementation of unconventional, non-IEEE-compliant arithmetics. Some of the hardware units in Table 1, for instance, use bfloat16, a 16-bit format that allocates 8 bits to the significand (including the implicit bit)

and 8 bits to the exponent and does not support subnormal numbers (*Intel Corporation, 2018*; *Intel Corporation, 2020*). It is worth noting that Volta is the first NVIDIA microarchitecture supporting tensor cores—the older Pascal GPUs, such as the P100, which are not reported in Table 1, supported binary16 arithmetic but did not include tensor cores. The NVIDIA Ampere A100 GPUs introduce a new 19-bit format called TensorFloat-32, which was designed to have the same dynamic range as binary32 (8 bits) and the same precision as binary16 (11 fraction bits, including the implicit bit) (*NVIDIA, 2020b*). In order to better understand the differences between the results computed using different systems, it is necessary to develop techniques to probe the numerical features of these units.

The situation is reminiscent of that before the widespread adoption of the IEEE 754-1985 standard, when different floating-point arithmetics had different properties. To address the issue, *Kahan (1981)* developed a program called paranoia that analyzes and diagnoses a floating-point arithmetic, which was subsequently translated into C by *Karpinski (1985)*. We follow a similar route: using idiosyncrasies of floating-point arithmetic, we design tests to better understand the numerical behavior of tensor cores, extending the testing approach recently introduced by *Hickmann & Bradford (2019)*. Our aim is to explore the following questions.

- Are subnormal inputs supported or are they flushed to zero? Can tensor cores produce subnormal numbers?
- Are the multiplications in (2) exact and the additions performed in binary32 arithmetic, resulting in four rounding errors for each element of $D$? In what order are the four additions in (2) performed?
- What rounding mode is used in (2)?
- Is the result of each floating-point operation in (2) normalized, or do tensor cores only normalize the final sum? What rounding mode is used for the normalization?

The answers to these questions are of wide interest because these accelerators, despite being introduced to accelerate the training of deep neural networks (*NVIDIA, 2017*, p. 12), are increasingly being used in general-purpose scientific computing, where their fast low precision arithmetic can be exploited in mixed-precision algorithms (*Abdelfattah et al., 2020*), for example in iterative refinement for linear systems (*Haidar et al., 2018a*, *2018b*, *2020*).

The results discussed here were produced by running our test suite, which is freely available on GitHub (https://github.com/mfasi/tensor-cores-numerical-behavior). The file `tc_test_numerics.cu` can be compiled following the instructions in the `README.md` file, so as to produce an executable that will run all the tests we describe in the following sections and produce a report. We run the test suite on an NVIDIA Tesla V100 SXM2 16GB GPU (Volta microarchitecture), an NVIDIA Tesla T4 16GB GPU (Turing microarchitecture), and an NVIDIA Ampere A100-PCIA 40GB GPU. We used version 10.1 of the CUDA library on the machines equipped with the Volta and Turing cards, and version 11.1 on the machines equipped with the A100 GPU, as GPUs based on the Ampere microarchitecture require at least version 11 of the CUDA library in order to utilize the

latest numerical features such as the TensorFloat-32 numerical type. Some graphic cards designed for intensive graphic processing workloads such as video gaming, computer-aided design, or computer-generated imagery, also include tensor cores: all the GPUs in the GeForce 20 and Quadro RTX series are based on the Turing microarchitecture, and those in the recently announced GeForce 30 series are based on the Ampere microarchitecture. We do not consider this wealth of different graphic cards here, as our focus is on the NVIDIA hardware that is present in the supercomputers in the TOP500 list and targets scientific computing applications. We stress, however, that the ideas we employ are very general and can be exploited to understand the numerical features of any hardware accelerator based on operations of the form (2).

Finally, we remark that the binary16 arithmetic implemented in the NVIDIA CUDA cores is not fully IEEE 754 compliant, as round-to-nearest is the only rounding mode implemented for elementary arithmetic operations (*NVIDIA, 2020c*)—we use this observation in "Support for Subnormal Numbers".

## PREVIOUS WORK

From a hardware perspective, instruction-level details, register configuration, and memory layout of the tensor cores in the NVIDIA Volta (*Jia et al., 2018b*; *Yan, Wang & Chu, 2020*) and Turing (*Yan, Wang & Chu, 2020*; *Jia et al., 2018a*) GPUs have been extensively described. Another study by *Basso, Dos Santos & Rech (2020)* explores the reliability of tensor cores in terms of rate of hardware errors in matrix multiplications. The main finding is that low-precision operations and usage of tensor cores increase the amount of correct data produced by the GPU, despite increasing the impact of numerical errors due to the use of lower-precision data. In order to quantify the accuracy of tensor cores, *Blanchard et al. (2020)* provide a rounding error analysis of what they call a block fused multiply-add (FMA), a generalization of the multiply-accumulate operation in (1) in which the matrix sizes, the precisions of the arguments, and the internal precision of the accumulator are taken as parameters.

*Markidis et al. (2018)* discuss various aspects of tensor cores and propose a technique, called precision refinement, to enhance the accuracy of mixed-precision matrix multiplication. Improving the accuracy of tensor-core-based matrix multiplications was further explored by *Mukunoki et al. (2020)*.

None of the these sources, however, examines to what extent tensor cores conform to the IEEE 754 standard or investigates how tensor cores compare with a matrix multiply-accumulate operation based on dot products implemented in software. *Hickmann & Bradford (2019)* explore some details of the numerical behavior of tensor cores with the main goal of inferring the hardware-level design of these units. Our work follows a similar approach and complements their findings by supplying further insights into the subject. We show that the additions in (2) are performed starting from the operand that is largest in magnitude, that at least 2 extra bits are used for carries, and that (2) may be non-monotonic for certain sets of inputs. Furthermore, we consider the second and third generation of the tensor cores, which equip the Turing T4 and Ampere A100 GPUs, respectively, and conclude that their internal accumulator has an extra bottom bit

**Table 2 Summary of the subsections of "Results".**

| Section | Devices used and description of tests performed on them |
|---|---|
| NVIDIA Volta Tensor Cores | Tests performed on the NVIDIA V100 (Volta) GPU |
| Support for subnormal numbers | Support for subnormal numbers (on the inputs, outputs and the computation of subnormals from the normalized inputs) |
| Accuracy of the dot products | Accuracy of the inner products performed as part of matrix multiply-accumulate (accuracy of scalar multiplies, accuracy of addition, and number of rounding errors introduced) |
| Rounding modes in tensor core computations | Tests for determining what rounding modes are used in the inner products and the final rounding |
| Features of the accumulator | Tests that explore the number of extra bits in the alignment step of floating-point addition inside the inner product (extra bits at the bottom of the internal significand) |
| | A test to find out whether the normalization is performed only at the end or after each addition in the computation of the inner products |
| | A similar test for the normalization in subtraction |
| | Tests for determining the number of extra bits for carries in floating-point addition (extra bits at the top of the internal significand) |
| | A test to check the monotonicity of the inner products |
| NVIDIA Turing tensor cores | All tests from "NVIDIA Volta Tensor Cores" rerun on the NVIDIA T4 (Turing) GPU |
| NVIDIA Ampere tensor cores | All tests from "NVIDIA Volta Tensor Cores" rerun on the NVIDIA A100 (Ampere) GPU |

compared with the tensor cores on V100 GPUs. Finally we make our test suite freely available, in order to guarantee reproducibility and facilitate testing other matrix multiply-accumulate units, such as the third generation tensor cores in the latest NVIDIA A100 GPUs (*NVIDIA, 2020b*).

# RESULTS

In this section we describe our testing methodology and give the results obtained on the three NVIDIA graphics cards we considered. Table 2 summarizes the tests we performed and indicates the subsections in which they are described. Our methodology is to find test cases that demonstrate specific numerical behaviors and, by making the (quite reasonable) assumption that the hardware has a consistent behavior across the input space, conclude that the discovered features should be true for all possible inputs.

## NVIDIA Volta tensor cores

Tensor cores can be accessed using the cuBLAS library, or the native hardware assembly instructions `HMMA.884` (*Jia et al., 2018a*, *2018b*) and `HMMA.1688` (*Yan, Wang & Chu, 2020*; *Jia et al., 2018a*). In our experiments, we opted for the warp-level C++ function `wmma::mma_sync()`, which performs a $16 \times 16 \times 16$ matrix multiply-accumulate operation. This is the lowest level interface to access the tensor cores in the NVIDIA CUDA programing environment. In order to use only a single tensor core, we set all but the top left $4 \times 4$ blocks to 0. We ensure that our experiments do use the tensor cores by running our test suite with the NVIDIA profiler `nvprof`, which shows the utilization levels of different hardware components on the GPU, and by observing that the assembly code produced by the `nvcc` compiler contains `HMMA` instructions. At the software level, tensor cores can be used in either binary16 or binary32 mode, which defines the number format of $D$ in (1).

**Table 3 Parameters of various floating-point formats: number of digits of precision including the implicit bit ($p$), boundary of the exponent range ($e_{min}$ and $e_{max}$), machine epsilon ($\varepsilon$), and smallest positive representable normal ($f_{min}$) and subnormal ($s_{min}$) numbers.** The formats from the IEEE 754 standard (*IEEE, 2019*) are binary16, binary32, and binary64.

|  | binary16 | bfloat16 | TensorFloat-32 | binary32 | binary64 |
|---|---|---|---|---|---|
| $p$ | 11 | 8 | 11 | 24 | 53 |
| $e_{max}$ | 15 | 127 | 127 | 127 | 1,023 |
| $e_{min}$ | −14 | −126 | −126 | −126 | −1,022 |
| $\varepsilon$ | $2^{-10}$ | $2^{-7}$ | $2^{-10}$ | $2^{-23}$ | $2^{-52}$ |
| $f_{min}$ | $2^{-14}$ | $2^{-126}$ | $2^{-126}$ | $2^{-126}$ | $2^{-1022}$ |
| $s_{min}$ | $2^{-24}$ | $2^{-133}$ | $2^{-136}$ | $2^{-49}$ | $2^{-1074}$ |

The experiments in this section were run on an NVIDIA Tesla V100 SXM2 16GB (Volta microarchitecture) GPU.

### Support for subnormal numbers

We start by investigating the support for subnormal numbers, as this knowledge will dictate what range of input values we are allowed to use in subsequent tests.

Table 3 compares precision, exponent range, machine epsilon, and magnitude of smallest representable subnormal and normal number for various floating-point number formats, including those supported by the three generations of tensor cores. By looking at the data in the table, we can make two important observations. First, conversion from binary16 to binary32 does not result in subnormal numbers. Second, the product of two binary16 numbers requires at most 22 bits for the significand, 6 bits for the exponent and one for the sign, and thus can be represented exactly in binary32.

As for tensor cores, there are multiple questions regarding the support of subnormal numbers.

1. Can tensor cores take binary16 subnormal numbers as inputs for $A$ and $B$ in (2) without flushing them to zero, use them in computation, and return binary16 or binary32 normal or subnormal results?
2. Can tensor cores take binary32 subnormal numbers as inputs for $C$ in (2) without flushing them to zero, use them in computation, and return subnormal binary32 results?
3. Can tensor cores compute subnormal numbers from normal numbers and return them?

The first question can easily be addressed by considering (2) and setting $a_{11} = 2^{-24}$, $b_{11} = 2^2$ (arbitrarily chosen), and the remaining elements to zero. The tensor cores return the subnormal result $a_{11} b_{11} = 2^{-22}$ in both binary16 and binary32 mode, thereby answering the first question in the affirmative.

An analogous idea can be used to clarify the second point: setting $c_{11}$ to the smallest positive binary32 subnormal $2^{-149}$ and all the elements of $A$ and $B$ to zero yields $d_{11} = 2^{-149}$, which confirms that the subnormal number $c_{11}$ is not altered by the dot product in (2). We note, however, that whether support for binary32 subnormals is needed is questionable. The absolute value of the smallest nonzero value that can be produced from the multiplication of two binary16 numbers is $2^{-48}$, thus $c_{11}$ would not contribute to

the sum if it were a binary32 subnormal: in binary32 arithmetic with round-to-nearest, one has that $2^{-48} + x > 2^{-48}$ only if $x > 2^{-48} \cdot 2^{-24} = 2^{-72}$, which is normal in binary32.

For the third question, we can obtain subnormal numbers as outputs in several ways. For instance, we can set $a_{11}$ to $2^{-14}$, the smallest normal number in binary16, and $b_{11}$ to $2^{-1}$, and confirm that tensor cores return the binary16 subnormal $2^{-15}$ in both binary16 and binary32 modes. Another possibility is to set $a_{11} = 2^{-14}$, $b_{11} = 1$, and $c_{11} = -2^{-15}$, which produces the subnormal binary16 number $d_{11} = 2^{-15}$. As mentioned above, it is not possible to obtain subnormal binary32 numbers from binary16 inputs in (2). In summary, these experiments demonstrate that there is full support for subnormal inputs in tensor cores.

One might wonder whether tensor cores natively support subnormals or some degree of software interaction is present. The NVIDIA profiler confirms that the experiments discussed in this section make use of the tensor cores, but we implemented an additional test to further reinforce the evidence that subnormals are supported in hardware. In "Rounding Modes in Tensor Core Computations" we show that tensor cores use round-towards-zero. We can use the fact that CUDA cores provide only round-to-nearest for binary16 computations to show that subnormals are in fact manipulated with tensor cores. In order to do so, we set $a_{11}$ and $a_{12}$ to 1, $b_{11}$ to the binary16 subnormal $2^{-23} + 2^{-24}$, $b_{21}$ to 2 and the other elements of $A$ and $B$ to 0. Since the addition in (2) is done in binary32 arithmetic, the smallest value that can be exactly added to $b_{21} = 2$ is $2^{-22}$. In this case, $b_{11} = 2^{-23} + 2^{-24} = \frac{3}{4} \cdot 2^{-22}$ will either be round down to 0, if round-towards-zero is being used, or rounded up to $2^{-22}$, if the summation is carried out using the CUDA cores, which support only round-to-nearest. This computation returned the value 2, meaning that $b_{11}$ was rounded down—a further indication that subnormals are natively supported by tensor cores.

### Accuracy of the dot products

Our second goal is to test the accuracy of the dot product (2) with the tensor cores. The first step is to check that the products of two binary16 values are computed exactly, which implies that the products must be kept in some wider intermediate format and accumulated without being rounded back to binary16. Specifically we want to test that $a_{1i}b_{i1}$, for $i = 1,\ldots, 4$, is computed exactly. This can be achieved by ensuring that the four multiplications produce floating-point numbers that are not representable in binary16 and checking that these are preserved and returned as binary32 entries of $D$.

In order to demonstrate this, we set the first row of $A$ and the first column of $B$ to $1 - 2^{-11}$ and $c_{11}$ to 0. The exact value of each partial product is $(1 - 2^{-11}) \cdot (1 - 2^{-11}) = 1 - 2^{-10} + 2^{-22}$, which, depending on the rounding mode, would be rounded to either $1 - 2^{-10}$ or $1 - 2^{-11}$, if the products were stored in binary16. As tensor cores produce the exact binary32 answer $d_{11} = 4 \cdot (1 - 2^{-10} + 2^{-22})$, we conclude that partial products are held exactly.

Another question is whether the precision of the 5-operand addition in (2) changes in any way when binary16 is used to store the elements of the matrices $C$ and $D$ in (1). The test is to set $a_{11} = b_{11} = a_{12} = 1 - 2^{-11}$, $b_{21} = 2^{-11}$, and the remaining elements to 0. In this test, the first product $a_{11}b_{11} = 1 - 2^{-10} + 2^{-22}$ requires precision higher than binary16 to be represented, whereas the second evaluates to $a_{12}b_{21} = 2^{-11} - 2^{-22}$. The sum

of these two products is $a_{11}b_{11} + a_{12}b_{21} = 1 - 2^{-10} + 2^{-11}$, which is representable in binary16 but would not be computed exactly by a binary16 accumulator, since storing the first product requires higher precision. Indeed we found that tensor cores output the exact value, confirming that the partial products are still held exactly even when $C$ and $D$ are in binary16 format.

A third question concerns the number of rounding errors in the 5-operand adder that accumulates the partial products. The dot product in (2) contains four additions, three to sum up the exact partial products and a fourth to add the binary32 argument $c_{11}$. Our expectation is that the additions are done in binary32 rather than exactly, as indicated by *NVIDIA (2017, 2020a)*. In order to confirm this, we can set the first row of $A$ to 1, thereby reducing (2) to

$$d_{11} = b_{11} + b_{21} + b_{31} + b_{41} + c_{11}, \tag{3}$$

and then run 5 different cases with one of the addends in (3) set to 1 and the rest set to $2^{-24}$. In this test, an exact addition would return $1 + 2^{-22}$, whereas inexact binary32 arithmetic would cause 4 round-off errors when adding $2^{-24}$ to 1, causing the number 1 to be returned. All permutations return $d_{11} = 1$, leading to the following conclusions.

- In the worst case each element of $D$ includes four rounding errors, which conforms to the block FMA model used by *Blanchard et al. (2020*, Sec. 2.1*)*.
- The partial products in (2) are not accumulated in a fixed order, but always starting from the largest value in magnitude. This sorting is necessary in order to know which arguments require to be shifted right in the significand alignment step of a standard floating-point addition algorithm (*Muller et al., 2018*, Sec. 7.3), (*Tenca, 2009*), and is most likely done in hardware. This is in line with the literature on hardware dot products (*Kim & Kim, 2009*; *Tao et al., 2013*; *Sohn & Swartzlander, 2016*; *Kaul et al., 2019*), where either sorting or a search for the maximum exponent is performed. Furthermore, this experiment demonstrates that none of the additions are performed before aligning the significands relative to the largest exponent: if evaluated before the arguments are shifted right relative to the largest magnitude arguments' exponent (by having multiple alignment stages), any other sum would evaluate to $2^{-24} + 2^{-24} = 2^{-23}$, a value that then could be added exactly to the total sum as the least significand bits would not be lost in the alignment.

In summary, each entry of $D$ in (1) can have up to four rounding errors, and the 5-operand additions that compute each element are performed starting from the largest summand in absolute value.

### Rounding modes in tensor core computations

If binary32 mode is used, only the four additions in (2) can be subject to rounding errors. The IEEE 754 standard defines four rounding modes for elementary arithmetic operations (*IEEE, 2019*, Sec. 4.3): round-to-nearest with even-on-ties, round-towards-zero, round-towards-minus-infinity, and round-towards-plus-infinity. In this section we use the

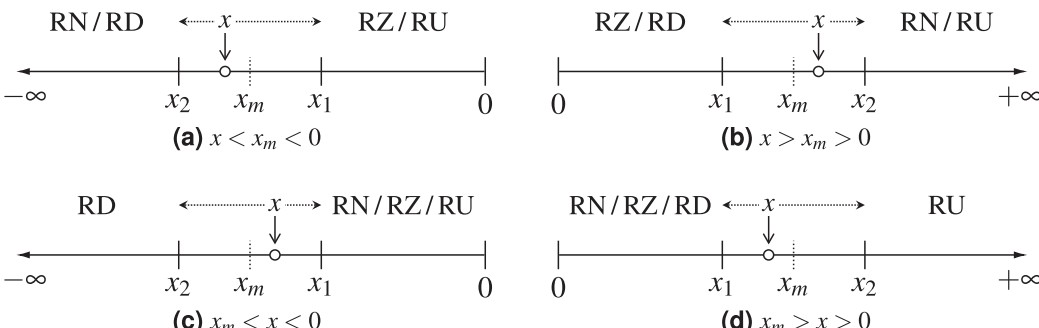

**Figure 1 Demonstration of the possible IEEE 754 rounding modes for different positions of the exact value $x$; $x_1$ and $x_2$ are the two floating-point numbers closest to $x$, and $x_m$ is the half-way point between them.** The dotted arrows surrounding $x$ show the direction in which various rounding modes would round it. (A) Negative axis, $x$ on the left of the middle point; (B) positive axis, $x$ on the right of the middle point; (C) negative axis, $x$ on the right of the middle point; (D) positive axis, $x$ on the left of the middle point.

notation defined by *Muller et al. (2018*, Sec. 2.2.1) and denote these four rounding operators by RN, RZ, RD, and RU, respectively.

As round-to-nearest is the default rounding mode in the IEEE 754 standard, we start by testing whether this is the rounding mode used by the tensor cores. This can be verified by setting any two partial products to values such that one of them would be rounded up only if round-to-nearest or round-towards-plus-infinity were used. If the result is rounded down we can conclude that the tensor cores implement either round-towards-zero or round-towards-minus-infinity, but neither round-to-nearest nor round-towards-plus-infinity (Fig. 1B). One such test is to set in (3) $b_{11} = 2$, $b_{21} = 2^{-23} + 2^{-24}$, and the remaining entries in the first column of $B$ to 0. Note that in binary32 arithmetic $\mathrm{RN}(2+x) > 2$ if $x > 2 \cdot 2^{-24} = 2^{-23}$, whereas the smallest positive $y$ such that $\mathrm{RZ}(2+y) > 2$ is $2 \cdot 2^{-23} = 2^{-22}$. The choice $b_{21} = \frac{3}{4} \cdot 2^{-22}$ is such that $x < b_{21} < y$, thus $\mathrm{RN}(b_{11}+b_{21}) = \mathrm{RU}(b_{11}+b_{21}) = 2 + 2^{-22}$ while $\mathrm{RZ}(b_{11}+b_{21}) = \mathrm{RD}(b_{11}+b_{21}) = 2$. Running this experiment on tensor cores returns $c_{11} = 2$, suggesting that either round-towards-zero or round-towards-minus-infinity is used for the additions in (2).

We can discriminate between these two rounding modes by repeating the same experiment on the negative semiaxis (Fig. 1A), which can be achieved by changing the sign of the nonzero elements in $B$. This experiment produces $c_{11} = -2$, and assuming that the rounding mode does not depend on the input, we conclude that the additions in (2) are performed in round-towards-zero. We note that this rounding mode is known to be the cheapest option to implement (*Santoro, Bewick & Horowitz, 1989*, Sec. 6.1) and is usually chosen for that reason.

When tensor cores are used in binary16 mode, the result computed in the format of the internal accumulator of the 5-operand adder has to be rounded to binary16 before being returned. In order to test the rounding mode used by this operation, we set $a_{11} = a_{12} = 2^{-24}$, $b_{11} = 2^{-1}$, $b_{21} = 2^{-2}$, and the rest of elements of $A$ and $B$ as well as $c_{11}$ to 0. The exact result of the dot product in this case is $2^{-25} + 2^{-26}$, which is not representable in binary16, and therefore will cause rounding errors in the result. Note that $2^{-25} + 2^{-26} = \frac{3}{4} \cdot 2^{-24}$, therefore

RN($2^{-25}+2^{-26}$) = RU($2^{-25}+2^{-26}$) = $2^{-24}$ while RZ($2^{-25}+2^{-26}$) = RD($2^{-25}+2^{-26}$) = 0. The fact that tensor cores return $2^{-24}$ confirms that round-towards-zero is not used, thereby suggesting that this conversion is performed in software rather than inside the tensor cores, which use round-towards-zero as we have determined above.

Figures 1C and 1D show how round-towards-minus-infinity and round-towards-plus-infinity could alternatively be determined by choosing $x$ such that $|x| < |x_m|$.

### Features of the accumulator

We now discuss some tests that allowed to determine various features of the internal accumulator of the 5-operand adder calculating (2). The quotes from NVIDIA that we provided in "Introduction" indicate that the internal accumulation is done in binary32 format; here we show that the internal format used by the accumulator has higher precision and that the partial sums are not normalized.

*Extra bits in the alignment of significands*
In order to compute the sum of two floating-point values, the floating-point adder matches the exponents of the two summands by shifting the significand of the number that has the smaller exponent to the right. In general this operation causes loss of information, as the least significant bits of the shifted significand are typically discarded, but it is customary to retain a few of these digits to guard the computation against numerical cancelation and to obtain correct rounding in round-to-nearest, round-towards-plus-infinity, and round-towards-minus-infinity. As tensor cores use truncation in the additions, we know that they do not require any such guard digits for rounding, and we can easily show that in fact they do not implement guard digits at all. If in (3) we set $b_{11} = 1$ and $c_{11} = -1 + 2^{-24}$, the tensor cores return $d_{11} = 2^{-23}$, which represents a relative error of ($2^{-23} - 2^{-24}$)/$2^{-24} = 1$.

*Normalization in addition*
When two floating-point values are added, the significand of the result may become larger than 2 (with $m > 2^p - 1$, where $m$ is an integer significand as in our definitions in "Introduction"), in which case a normalization step (shifting the significand right by one place and increasing the exponent by one) is required (*Muller et al., 2018*, Sec. 7.3). In an IEEE 754-compliant arithmetic, the result of each partial sum in (2) would be normalized, as floating-point adders always normalize the result in order to produce a normalized floating-point number. But tensor cores compute the whole expression (2) in hardware rather than by means of IEEE 754 elementary arithmetic operations, and it is natural to ask whether each partial result in (2) is normalized or only the final answer is. We can verify this by adding values chosen so to produce different results with and without normalization of the partial sums. In (3) we set $c_{11} = 1 - 2^{-24}$ and the elements of the first column of $B$ to $2^{-24}$.

Recalling that the values are accumulated on the summand of largest magnitude, we start by examining what would happen if each partial result were normalized. The exact value of the partial sum $s = c_{11} + 2^{-24}$ is 1, and the normalization step would shift the significand by one bit to the right. At this point the three remaining addends would be smaller than the least significant bit of the partial sum, thus adding them to $s$ separately

would have no effect with round-towards-zero. If the partial results were not normalized, on the other hand, the sum of $c_{11}$ and $2^{-24}$ would be held with one extra bit, and the remaining addends could be added to it. Running this test on tensor cores shows that only the final result of the dot product is normalized. This has probably been done in order to simplify the implementation; a similar choice was made for example in the hardware accelerator for performing vector inner product described by *Kim & Kim (2009)*.

*Normalization in subtraction*
As the products in (2) can be positive as well as negative, some of the partial sums can in fact be subtractions. The significand of the difference of two floating-point numbers may be smaller than 1, in which case the result has to be normalized by shifting the significand left and decreasing the exponents accordingly until the result becomes a normal number. We can show that tensor cores do not perform this kind of normalization as follows. If in (3) we set $c_{11} = -1 + 2^{-24}$, and two of the elements of the first column of $B$ to 1 and $-2^{-24}$, we have that $d_{11}$ evaluates to 0 if the partial sums are normalized. Instead, running this experiment on tensor cores yields $d_{11} = 2^{-23}$, which can be explained as follows. When the sum is evaluated as $(1 + c_{11}) - 2^{-24}$, then the lack of guard digit implies that $1 + c_{11}$ evaluates to $2^{-23}$, and if the partial results were normalized the tensor cores would return $2^{-23} - 2^{-24}$, which can be represented exactly in binary32 format's precision. If, on the other hand, the sum were evaluated as $(-2^{-24} + 1) + c_{11}$, the first sum would return 1 due to the lack of guard digit, and the lack of normalization would not have any effect in this case. We can conclude that the result of the subtraction is not normalized, as long as we assume that the summands in (3) are accumulated on the largest in magnitude in a fixed order.

*Extra bits for carry out*
Another question concerns the number of extra bits required due to lack of normalization. If only the final result is normalized, then accumulating $k$ addends requires $\lceil \log_2 k \rceil$ bits for the carry-out bits (*Ercegovac & Lang, 2004*, Sec. 3.1), and the hardware for accumulating the 5 values in (2) would internally require $\lceil \log_2 5 \rceil = 3$ extra carry-out bits at the top of the significand. We can prove that the internal accumulator of the 5-operand adder in tensor cores has at least two extra bits as follows. In (3) we take $c_{11} = 1 + 2^{-22} + 2^{-23}$, which sets the two least significant bits of the significand to 1, and assign to the first column of $B$ a permutation of the values 1, 1, 1, and $2^{-23}$. We consider all four possible permutations of these values in the first column of $B$, as we assume that the addends apart from the largest in magnitude are not sorted. The main idea is to show that if 1 is added to $c_{11}$ three times, then the last two bits of $c_{11}$ are not dropped as they would be if the accumulation were performed using IEEE 754 floating-point arithmetic, as the significand of $c_{11} + 3 > 4$ would have to be shifted by two places to the right in order to be normalized. Then, when $2^{-23}$ is added at the end, the carry propagates into the third bit from the bottom and therefore is not lost in the final normalization step. If there are 2 extra bits, then all the four possible orderings of the first column of $B$ will return the exact result $4 + 2^{-21}$. Running these tests on tensor cores, we found that all four combinations returned the

exact result, thereby proving that the significand of the internal accumulator has at least 2 extra bits for carries.

This technique cannot be used to incontrovertibly show that the 5-operand adder has a third extra bit for carries. On the one hand, all inputs to the multi-operand adder must have the most significant bit of the fraction (the implicit bit) set to 1, in order to produce carry that requires all three extra bits, on the other, the result of one of the four products of the form $a_{1k}b_{k1}$ in (2) must have the least significant bit of the fraction set to 1. As the product of two binary16 numbers can have at most 22 significant digits, no combination of input can produce a partial product with the required characteristics.

It is possible, however, to show the presence of the third bit if we assume that the alignment of the significand is always performed so that the most significant bit of the largest summand in the 5-operand adder occupies the left-most position. Since we know that there are two extra bits and no normalization, we can show that there is also a third bit by showing that there is no overflow when each of the four additions in (3) causes carry out.

We can set, for example, $c_{11} = 1 + 2^{-1} + 2^{-2} + 2^{-3}$, $b_{11} = 1$, $b_{21} = 1 + 2^{-1}$, $b_{31} = 1 + 2^{-1} + 2^{-2}$, and $b_{41} = 1 + 2^{-1} + 2^{-2} + 2^{-3}$ and observe that the tensor core returns $d_{11} = 8$. If only two extra bits were present, on the other hand, overflow would occur and the adder would incorrectly return $d_{11} = 0$. In summary, we can conclude that the internal significand of the tensor cores is most likely 27 bits wide in Volta cards and 28 bits wide in Turing and Ampere cards (see "NVIDIA Turing tensor cores" and "NVIDIA Ampere tensor cores").

It is worth noting at this point that if (1) there is no normalization, (2) the additions in (2) start with the largest value in magnitude, and (3) all of the significands of the addends are shifted right relative to the exponent of the largest value in magnitude, then the order in which the remaining addends are accumulated will not impact the final result.

In the test case above, by replacing one of the $2^{-23}$ by 1 we can also confirm, using the methods developed in "Rounding Modes in Tensor Core Computations", that the rounding mode in the final normalization step (internal accumulator conversion to binary32 answer) is round-towards-zero.

## Monotonicity of dot product

The observation in the previous section raises one final question regarding the monotonicity of the sums in (2). The accumulation is monotonic if in floating-point arithmetic the sum $x_1 + \cdots + x_n$ is no larger than $y_1 + \cdots + y_n$ if $x_i \leq y_i$ for all $1 \leq i \leq n$.

We can show that the lack of normalization causes the dot product in tensor cores—and most likely in any other similar architectures in which partial sums are not normalized (*Kim & Kim, 2009*; *Tao et al., 2013*; *Sohn & Swartzlander, 2016*; *Kaul et al., 2019*)—to behave non-monotonically. Let us consider (3) and set all the elements in the first column of $B$ to $2^{-24}$ and then $c_{11}$ to $1 - 2^{-24}$ and 1 in turn. When $c_{11} = 1 - 2^{-24}$, the difference in exponents guarantees that the values in $B$ are large enough to be added to $c_{11}$. This causes the result to become larger than 1, requiring a normalization that returns $1 - 2^{-24} + 3 \cdot 2^{-24} = 1 + 2^{-23}$. On the other hand, when $c_{11} = 1$, none of the summands in (3) is large enough to be added to $c_{11}$, as the elements in the first column of $B$ are all zeroed out

during the significand alignment step of each addition. This happens because the exponent of 1 is larger than that of $1 - 2^{-24}$. In summary we have

$$d_{11} = c_{11} + 2^{-24} + 2^{-24} + 2^{-24} + 2^{-24} = 1 + 2^{-23} \quad \text{when } c_{11} = 1 - 2^{-24},$$

$$d_{11} = c_{11} + 2^{-24} + 2^{-24} + 2^{-24} + 2^{-24} = 1 \quad \text{when } c_{11} = 1.$$

These two sets of inputs demonstrate that tensor cores can produce non-monotonic behavior.

### NVIDIA Turing tensor cores

NVIDIA Turing T4 GPUs are equipped with the second generation of tensor cores, which adds an integer matrix multiply-accumulate operation. It is not documented whether the binary16/binary32 tensor core arithmetic in Turing chips differs from that in Volta cards, therefore it is of interest to run the test suite we designed on one of the Turing cards.

We ran all the above experiments on an NVIDIA Tesla T4 16GB (Turing microarchitecture) GPU, and noticed that some of the results were different from those obtained on a V100 GPU. We found that this is due to the presence of an additional extra bit of precision at the bottom of the significand of the internal accumulator of the 5-operand adder. This has an impact over several of the tests above: the operation $1 + 2^{-24} + 2^{-24}$, for example, can now be performed exactly because of the presence of the extra bit. The results obtained on the V100 GPU can be replicated by means of a suitable change of the constants that are chosen depending on the number of extra bits in the accumulator. For instance, in the test for the order of operations in "Accuracy of the Dot Products", the constant $2^{-24}$ should be replaced by $2^{-25}$, which is the value of the next bit to the right. Using this approach, we found that all the conclusions we drew about the tensor cores in V100 GPUs remain valid for the second version of tensor cores in the T4 GPUs, with the only exception of the extra bit at the bottom of the internal storage of the 5-operand adder.

If we denote the fixed-point format with $I$ integer bits and $F$ fraction bits by $\{I.F\}$, the significand of the internal format of the 5-operand adder of a V100 GPU has format $\{3.23\}$ (or $\{4.23\}$ if 3 extra bits for carries are present as discussed in "Features of the Accumulator"), whereas that of a T4 GPU has format $\{3.24\}$ (or $\{4.24\}$). The final normalization and rounding produce a number whose significand has format $\{1.23\}$, which is the format of the significand of a binary32 floating-point number.

### NVIDIA Ampere tensor cores

As shown in Table 1, the Ampere microarchitecture offers four different variants of tensor core operations (input/output format): binary16/binary32, bfloat16/binary32, TensorFloat-32/binary32, and binary64/binary64. In this section we summarize, for each of these four configurations, the results obtained by running our test suite on the A100 GPUs. We refer to the 4-element dot product in (2), even though some of the tensor core operations available on A100 GPUs use input vectors of dimension two or eight, as can be seen in Table 1. In our tests we only use the first few elements of each input vector, so (2) is relevant even when dealing with operations that would require some of the input vectors to have eight elements, as we tacitly assume that all the remaining entries are set to 0.

In binary16/binary32 mode, we found no differences between the Ampere and Turing tensor cores, as all the tests in our suite produce the same results on the two GPUs. Therefore we moved to the other three modes, which are new as they were introduced with the NVIDIA Ampere architecture.

We started by adapting our test suite to the bfloat16/binary32 mode. The main difference in this configuration is that the subnormals in bfloat16 are a subset of the subnormals in binary32, whereas all binary16 subnormals were normalized values in binary32. For this reason, in binary16/binary32 mode, we could only observe that binary32 subnormals in the input are preserved in the output. In this mode, however, binary32 subnormal outputs may in principle be produced during the computation from normal inputs, and we can verify that this is indeed the case with the following experiment. If in (2) we set $a_{11} = 2^{-126}$, $b_{11} = 2^{-1}$ (normalized bfloat16 values), and the remaining coefficients to zero, the tensor cores correctly produce the subnormal binary32 result $d_{11} = 2^{-127}$. To summarize, this precision configuration presents the same numerical behavior as the binary16/binary32 configuration in the T4 GPU tensor cores, and an additional test for subnormals confirmed full support for bfloat16 subnormal input and binary32 subnormal output.

Next we looked at the A100 with tensor cores configured in TensorFloat-32/binary32 mode. TensorFloat-32 can represent more subnormal values than bfloat16 due to extra 3 bits in the significand (Table 3), and this was taken into account in our tests. The rest of the test suite was run with the same input used for the binary16/binary32 configuration, since the significands are of the same width, and produced the same results. Our conclusion is that the TensorFloat-32/binary32 mode exhibits the same features as the binary16/binary32 configuration in the T4 GPU.

The last configuration of the Ampere tensor cores is binary64/binary64. We found that the numerical behavior of this mode differs significantly from that of the other three configurations that are available on these graphic cards. In the multi-operand addition in (2), in particular, we observed that round-to-nearest with ties to even is used, and that the normalization is performed after each addition rather than only once at the end of the computation. As the result of each addition is normalized, no extra bits for carries are required, and the monotonicity issue discussed in "Features of the accumulator" does not occur for this rounding mode. We also note that since accumulation is performed in binary64, there would be no advantage in holding the binary64 products exactly. Moreover, the additions in (2) are not always performed starting from the largest addend, as was the case with the other modes. In summary, our results suggest that in binary64/binary64 mode the Ampere tensor cores have the same numerical features that a matrix multiply-accumulate operation implemented using IEEE 754-compliant arithmetic operations would have.

## CONCLUSIONS

Our experiments indicate that the tensor cores in the NVIDIA V100 architecture have the following numerical features. The first two of these (except the rounding mode for the second one) are stated in the NVIDIA documentation and are confirmed by our

experiments, while the rest are not documented by NVIDIA and have been revealed by our tests.

1. The binary16 products in (2) are computed exactly, and the results are kept in full precision and not rounded to binary16 after the multiplication.
2. The additions in (2) are performed using binary32 arithmetic with round-towards-zero.
3. Subnormal numbers in binary16 and binary32 are supported.
4. The five summands in (2) are accumulated starting with the largest in absolute value.
5. Only the final result of (2) is normalized; the partial sums are not, and the accumulator has an internal significand that is 27 bits wide to accommodate for carries of the 5-term addition.
6. The dot products in tensor cores are non-monotonic: in some cases, increasing the magnitude of the inputs to (2) reduces the magnitude of the final result, when the summands in (2) are all nonnegative or all nonpositive,

The same properties were found in the second generation tensor cores which equip the NVIDIA T4 GPUs, the main difference being one extra bit of precision in the significand of the internal accumulator of the binary32 5-operand adder. In the third generation of tensor cores, available on the NVIDIA A100 GPUs, all features were the same as in the T4, with the exception of the binary64/binary64 tensor cores, which we found are implemented to behave as if implemented in IEEE 754-compliant arithmetic.

The test suite that we have developed as part of this work can be used to test various properties of the floating-point arithmetic of future versions of tensor cores as well as similar accelerators. We aim to keep extending our test suite by adding new test cases for standard non-mixed-precision binary32 or binary64 dot product or matrix multiply units, as well as for integer arithmetic. The NVIDIA Turing and Ampere tensor cores, for instance, added support for 4- and 8-bit integer modes (*NVIDIA, 2018*, *2020b*), and rounding issues become relevant when these are used to implement fixed-point arithmetic.

## ACKNOWLEDGEMENTS

The authors thank the University of Manchester for providing access to the NVIDIA V100 graphic cards through the Computational Shared Facility, the IT services of the university for setting up access to the NVIDIA T4 graphic cards through the Amazon Web Services, and the Innovative Computing Laboratory at the University of Tennessee, Knoxville, US for providing access to the NVIDIA A100 graphics cards.

### Funding

Mantas Mikaitis was supported by an EPSRC Doctoral Prize Fellowship and grant EP/P020720/1. Nicholas J. Higham and Srikara Pranesh were supported by Engineering and Physical Sciences Research Council grant EP/P020720/1. Nicholas J. Higham and

Massimiliano Fasi were supported by the Royal Society. The funders had no role in study design, data collection and analysis, decision to publish, or preparation of the manuscript.

## Grant Disclosures

The following grant information was disclosed by the authors:
EPSRC: EP/P020720/1.
Engineering and Physical Sciences Research Council: EP/P020720/1.
Royal Society.

## Competing Interests

Nicholas J. Higham is an Academic Editor for PeerJ Computer Science.

## Author Contributions

- Massimiliano Fasi conceived and designed the experiments, performed the experiments, analyzed the data, performed the computation work, prepared figures and/or tables, authored or reviewed drafts of the paper, and approved the final draft.
- Nicholas J. Higham analyzed the data, prepared figures and/or tables, authored or reviewed drafts of the paper, and approved the final draft.
- Mantas Mikaitis conceived and designed the experiments, performed the experiments, analyzed the data, performed the computation work, prepared figures and/or tables, authored or reviewed drafts of the paper, and approved the final draft.
- Srikara Pranesh analyzed the data, prepared figures and/or tables, authored or reviewed drafts of the paper, and approved the final draft.

## Data Availability

The repository is available at GitHub and contains the source code used for performing the experiments: https://github.com/mfasi/tensor-cores-numerical-behavior.

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
