# Peer review of "Numerical behavior of NVIDIA tensor cores"

_PeerJ Computer Science, doi:10.7717/peerj-cs.330_

## Round 0.1 · original submission · Major Revisions

I reviewed the paper myself and analyzed the reviewers' comments. I concur with the detailed reviewers' comments.

Reviewer 1 ·

Basic reporting

Spotted typos:

(line 174) in a several ways

Experimental design

The paper presents the methods to obtain essential parameters of the floating
point arithmetic of bespoke floating-point hardware present in NVIDIA compute
cards in Tesla product group. The tested cards were V100 and T4 and the code
for tests was made publicly available. This allows the authors to circumvent
the lack of low-level details of hardware.

Validity of the findings

Briefly stated, what is available under the provided link and what requirements
must be met by the user system? Is a Github account required? CUDA versions
that are supported? And which CUDA versions support tensor32 format?

Additional comments

NVIDIA Pascal P100 supports fp16 but no tensor cores' arithmetic computational
model. Is that correct? Could this be clarified somewhere for Table 1? Also,
are the Geforce GTX cards such as 2060 and share the same hardware?

Table 2 has room for adding values mandated by IEEE for their formats. Would it
be possible to include some of them?

How is the implementation of tensor32 on the tested V100 differ from the same on
T4 and A100? Does the type represent the new standard put forth by NVIDIA or
each microarchitecture features a bespoke variant rendering each a unique type
with subtle differences?

How is NVIDIA profiler involved in confirming the use and probing the behavior
of tensor cores?

Reviewer 2 ·

Basic reporting

This paper does an interesting work of performing experiments which are not reported in the Tensor Cores documentation. The authors should have explained the definition of terms like Subnormal numbers, normalization in the Introduction Section as new readers may find it difficult to follow.

Experimental design

The design and choice of experiments were interesting. As Tensor Cores are designed to accelerate Matrix Multiplications, it will be interesting to perform an application like Convolutional Neural networks (Eg. Vgg16) or any other scientific computation instead of just (4*4*4) matrix multiplication with arbitrary values.

Validity of the findings

A table with a summary of Experiments A,B,C,D would be more appealing to the reader. As the experiments were performed on V100 Architecture, does it apply to other architectures like Turing and TPUs ?

Reviewer 3 ·

Basic reporting

1. + References, results, etc. sufficiently present and included in a coherent manner in the paper.
2. - The authors use "held exactly" in several places (e.g., 3.1.2) when they mean held as if it were a IEEE 754 single precision floating point computation (not an exact format).

Experimental design

1. - The authors do not specifically identify the knowledge gap relative to Hickmann and Bradford's work. For example, subnormal value handling and creation, addition schedule in (2) (see align and truncate discussion in Hickmann and Bradford), and monotonicity (follows from align and truncate) are present in prior works. The current work seems to only fill the gap of more explanation, open code, and application to a new GPU.
2. + Provide code appears to be sufficient to easily replicate author's work. Even if not, the descriptions within the paper are sufficient to replicate the work.
3. - Why did the authors not determine if there was a third carry bit for the 5-operand accumulation? Or was it confirmed that it didn't exist in the V100? (this may simply be a reporting issue)

Validity of the findings

1. - The assertions/conclusions of the findings should be phrased in such a way as to indicate these are "likely" numerical features since there were not exhaustive tests and there is no claim that a standard is adhered to or verified against.
2. - Certain assertions with regard to computation ordering/hardware are overly stated. For example, lines 230-234 "Furthmore, this exmperiment demonstrates that none of hte additions is performed in parallel with the first addition to the largest magnitude value..." Simply means that the alignment step of the additions all happens first, not that the actual adds are sequential (see Hickmann and Bradford's estimated design).

Additional comments

While the paper reads well in terms of explanation of logic and I appreciate the provided code, it is unclear what knowledge gap is being filled. When comparing point by point with Hickmann and Bradford I cannot find such a gap.

---

## Round 0.2 · Minor Revisions

Thanks for undertaking the revision. Once you complete the revision, the paper would be in acceptable form.

Reviewer 1 ·

Basic reporting

Thank you for addressing the issues from my review.

No further comments in this part.

Experimental design

Thank you for addressing the issues that expanded and clarified (for me) the experimental sections of the manuscript.

I have no additional comments in this part.

Validity of the findings

Thank you for responding to the suggestions wrt validity of findings.

No comments to add in this revision.

Additional comments

Thank you for the thorough review.

I reviewed the manuscript and have no further comments to report.

Reviewer 2 ·

Basic reporting

Thanks for adding the paragraph which define all the terms

Experimental design

no comment

Validity of the findings

The summery table seems to be fine.

Reviewer 3 ·

Basic reporting

No change.

Experimental design

From response letter:
13 a. The paper's results confirm Hickman and Bradford -- an integer adder tree (compressor or otherwise) will maintain all carries since there is no rounding/truncation. This is what would be referred to as a 24-bit datapath. If this were not the case it would not meet Nvidia's claim of "accumulating with at least single precision" and there would be large errors observed or internal rounding points. I still see no knowledge gap here. I would suggest using the phrasing you added "the internal significant of the tensor cores is most likely 27 bits wide in Volta cards and 28 bits wide in Turing cards" in your conclusion, etc. rather than "carry-out" bits.

b. While I do appreciate the logic of testing extreme values and the likelihood that hardware behaves in consistent manner, a single observation hardly refutes someone else's observation.

Validity of the findings

Rewording has addressed comments.

Additional comments

Overall, based on the responses and published tests, I support accepting this work with a suggestion to change in 13a.

---

## Round 0.3 · accepted · Accept

Thanks for the updated and new results included in the paper. It makes the results stronger. Thanks for your interest in the journal.